# Genetic Loci Underlying Awn Morphology in Barley

**DOI:** 10.3390/genes12101613

**Published:** 2021-10-14

**Authors:** Biguang Huang, Weiren Wu, Zonglie Hong

**Affiliations:** 1Key Laboratory for Genetics, Breeding and Multiple Utilization of Crops, Ministry of Education, Fujian Agriculture and Forestry University, Fuzhou 350002, China; hbg1989@163.com; 2Fujian Collegiate Key Laboratory of Applied Plant Genetics, Fujian Agriculture and Forestry University, Fuzhou 350002, China; 3Department of Plant Sciences, University of Idaho, Moscow, ID 83844, USA; 4Fujian Key Laboratory of Crop Breeding by Design, Fujian Agriculture and Forestry University, Fuzhou 350002, China

**Keywords:** awn, barley, morphology, gene mapping, genetic epistasis, pleiotropism

## Abstract

Barley awns are highly active in photosynthesis and account for 30–50% of grain weight in barley. They are diverse in length, ranging from long to awnless, and in shape from straight to hooded or crooked. Their diversity and importance have intrigued geneticists for several decades. A large collection of awnness mutants are available—over a dozen of them have been mapped on chromosomes and a few recently cloned. Different awnness genes interact with each other to produce diverse awn phenotypes. With the availability of the sequenced barley genome and application of new mapping and gene cloning strategies, it will now be possible to identify and clone more awnness genes. A better understanding of the genetic basis of awn diversity will greatly facilitate development of new barley cultivars with improved yield, adaptability and sustainability.

## 1. Awnness or Awnlessness

To be, or not to be, that is the question. With this soliloquy, William Shakespeare once pondered, through Hamlet’s words, upon a state of being or not being, a forever lasting philosophic question about life and death. A similar balance about awnness or awnlessness should have puzzled our human ancestors some 8000 years ago when barley (*Hordeum vulgare* L.) was domesticated. Wild grass species, including wild barley (*Hordeum spontaneum*), are mostly awned, whereas their cereal crop relatives vary. Domestication has led to the disappearance of awns from most popular cultivars of rice and wheat, while, on the contrary, most modern barley varieties still preserve awns. The awn on the barley spike has no nutritional value and can cause lacerations in the mouth of animals. One of the most important breeding goals for forage barley is the elimination of the awn without reducing the grain yield. A better understanding of the genetic basis for awnness is key to advancing breeding programs aimed at barley cultivars with special awn types. Although environmental conditions affect awn development to a certain degree, the question of ‘to awn or not to awn’ is largely answered by the genetic composition of the barley genome. The inheritance of the awn trait in barley has been a topic of genetic study for several decades, primarily due to the importance of the awn to grain yield and its hindrance in forage quality and palatability [1]. The specific objectives of this review are to provide an update of research work conducted by various groups in the field of awn development and inheritance over the last several decades, with a focus on the recent progress in genetic loci underlying awnness that have been identified, mapped or cloned in barley. The complexity of gene–gene interactions and pleotropic effects of awn loci are analyzed and the perspectives of future research work in this field are presented in this review.

In addition, the awn is not an essential organ for the survival and reproduction of barley, allowing awn shape to vary dramatically through natural selections and chemical mutagenesis. The awn protrudes from the canopy in fields and morphological differences are easy to score, leading to large collections of abundant germplasm resources of awn mutants. In addition, barley represents an excellent model for genetic research of crops: it is diploid with a low chromosome number (2*n* = 14), can be both inbred and outbred easily, and adapts to diverse climate conditions [2,3]. The completion of the barley genome sequencing project [4,5,6] has facilitated identification and cloning of agronomically important genes, such as the row-type locus *VRS3* [7,8].

## 2. Biological Function and Anatomy of Awns

The awn is a green floral organ extending from the lemma, which itself is a modified sepal in origin [9,10]. Although it is a nonessential organ to the growth and reproduction of the plant, when present, the awn is highly active in photosynthesis and can contribute significantly to the filling of developing grains [11]. In addition, the awn can protect against animal predation before grain harvest, help disperse dry seeds in the wild and anchor germinating seeds to the soil [11,12]. Awnless and short awn cultivars are likely to perform better under extreme conditions such as drought [12,13]. Microscopic analysis of barley awn transverse and longitudinal sections reveals that the green cells of awns are arranged into two long strands of chlorenchyma, which are placed in between the three vascular bundles (Figure 1) [11,12]. Barley awns are highly diverse in morphology, ranging from straight to hooded or crooked shapes [14]. The development of awns is controlled by a set of genetic loci [15], of which a few have recently been cloned. This review provides an account of these awn genes and their interactions at the genetic level.

## 3. Morphological Diversity of Barley Awns

The diversity and development of awns in barley have long been a topic of genetic research and now are subjects of intense molecular biology and genomic studies [14,16]. Based on the presence of awns, barley germplasms can be divided into awnless and awned types, with the latter subdivided further into hooded, leafy, straight and crooked phenotypes (Figure 2). The straight awns (single or triple) can be long, short or awnlet, while the hooded ones can be classified as either normal hooded, elevated hooded, or subjacent hooded. Furthermore, barley awns do not behave consistently in spikelets between the central and lateral rows of the same spike. For example, when the central row develops long awned spikelets, the lateral row can be short awned or even awnless. When both the central and lateral rows develop long awns, the awns in the lateral rows are generally a little shorter than those of the central rows. Hooded awns have their own unique structures, which can sometimes develop inverse, fertile flowers capable of producing seeds [14,17]. The striking inverse flower of hooded awns is thought to be the result of the transformation of the awn into reiterative inflorescence axes [18]. Subjacent hooded awns are similar to the hooded ones, but differ by bearing distal awns without an epiphyllous flower, reflecting differing degrees of phenotypic penetrance [19]. In anatomy, the awn is a transformed leaf blade, as illustrated in the *leafy lemma* mutant (Figure 2K). Barley germplasms from the Tibet plateau have been shown to exhibit highly diverse awn characteristics, ranging from long awns on central rows but long, short, or no awns at all on lateral rows. Similarly, other cultivars can have short awns on the central row with short or awnless on the lateral row, hooded awns on the central and hooded, short or awnless on the lateral, to completely awnless on both the central and lateral rows [20,21].

## 4. Cloned Awnness Genes in Barley

Three awnness-specific genes have so far been cloned in barley and they each encode a type of transcription factor (Table 1). The mutant *hooded lemma 1* (or *Kap1* for “*kapuze*” in German and “*hood*” in English) produces a dramatic hood-shaped awn which is attached to the lemma and may contain an extra flower of inverse polarity on the lemma (Figure 2N). The dominant *Kap1* locus has been cloned using a homologous cloning approach and encodes a *Knox3* transcription factor. *Kap1* is also referred to as *HvKnox3* or *Bkn3* for the barley *Knox3* gene. The *Kap1* mutant contains a duplication of 305 bp in intron IV of *Bkn3* [22]. Surprisingly, this insertion in intron IV does not change the exon sequences, but leads to a 2.5-fold increase in its mRNA level, possibly due to an enhanced splicing rate of intron IV in the *Kap1* mutant. This observation suggests that proper regulation of the expression level of the hooded gene *HvKnox3* is important. This conclusion is supported by evidence from overexpression of maize *knotted1* (*kn1*), the ortholog of barley *Kap1*, in transgenic barley, which phenocop the *Hooded* mutant awn appearance [23]. The KNOX-type transcriptional regulators, a family of homeodomain-containing proteins, are known to regulate the maintenance of the shoot apical meristem and the initiation of lateral organs in plants.

The second awnness gene cloned in barley is *vrs1*, on the locus which is also responsible for the six-rowed spike. Its wild-type allele *Vrs1*, responsible for the two-rowed spike, encodes a homeodomain- and leucine zipper motif-containing transcription factor [25]. *Vrs1* is expressed in the lateral-spikelet primordia and suppresses development of the lateral rows. Its loss-of-function alleles (*vrs1*) cannot suppress lateral row development, resulting in formation of fertile six-row spikelets. Mutant alleles of *vrs1* are found to contain missense mutations, deletions, alterations in splicing sites or mutations in the gene promoter [25]. *vrs1* is allelic to the *reduced lateral spikelet appendage* (*lr*) locus, characteristic of having normal and long awns on the central spikelets and awnlessness on the lateral spikelets [26,28,29].

The third cloned barley awnness gene, *Lks2*, encodes a SHORT INTERNODES (SHI)-type transcription factor [12]. The SHI family proteins are plant-specific and contain two conserved regions, the RING-finger motif and the IGGH domain, the former is implicated in zinc-binding whereas the latter is required for dimerization and transcription activation [30]. Of the four SHI-related genes in the barley genome, only *lks2* is known to display phenotypes of awns and pistils. *lks2* (for *length 2* short awns) is a spontaneous recessive mutation that reduces awn length by 50% and also causes sparse stigma hairs and reductions in awn thickness. The short awn trait is thought to have an adaptive advantage in humid growth conditions and is found in many landraces in Eastern Asia [31]. *lks2* is allelic to *unbranched style 4* (*ubs4*) and the short awn locus *breviaristatum-d* (*ari-d*). *Lks2* is highly expressed in awns and pistils, consistent with the defective mutant phenotypes [12]. Analysis of the interaction between *Kap1* (*K*) and *lks2* has revealed that the short awn gene *lks2* is recessive and epistatic to the hooded gene *K/k* [32].

In addition to the three cloned awnness-specific genes, the semi-dwarf gene *uzu1* (previous name *uzu* in Japanese and “swirl” in English) has also been shown to affect awn development [33]. The *uzu1* gene (or *HvBRI1*) encodes a brassinosteroid (BR)-receptor protein. In the *uzu1* mutant, a single-nucleotide A to G change at the position of 2612 of the *HvBRI1* gene results in the amino acid substitution of histidine to arginine [34]. The spontaneous mutant *uzu1* produces short awns of about 1/3 of normal length. However, this awn reduction in *uzu1* is not awn-specific, but is a result of collateral damage from the overall reduction in plant length caused by the defect in BR perception.

It is not surprising that the first three cloned barley awnness genes all encode transcription factors, because these kinds of proteins typically regulate the expression of a set of genes that have functions in a pathway or a developmental process, like awn initiation and extension. With the availability of the sequenced barley genome [4,5,6] and application of various genome sequencing-based cloning technologies [35], it is expected that more barley awnness genes will be cloned in the near future. Awnness genes with functions other than transcription factors may be uncovered, which may provide new insight into the mechanism of awn initiation and development.

## 5. Mapped Genetic Loci for Awns in Barley

A three-letter code for genes and the new *Triticeae* system for chromosome designation have been adopted in Table 1 and Table 2 to describe barley awnness loci [15]. Previous designations of barley chromosomes 1 through 7 correspond to 7H, 2H, 3H, 4H, 1H, 6H and 5H, respectively [36]. The 12 awnness loci identified so far (Table 1 and Table 2) underlie a spectrum of awn phenotypes, including awnlessness (*Lks1, Lsa1*), short awns (*lks2*, *lks5/lel2*, *lks6*, and *ari-a*), reduced lateral spikelet appendage (*lr*), leafy lemma (*lel1*), hooded lemma (*Kap1*), subjacent hooded lemma (*sbk1*), short, crooked awn (*sca1*), and triple awns (*trp1*). Their chromosome locations and their relative distances to known morphological marker genes are shown in Figure 3.

The *leafy lemma* (*lel*) mutants develop dysmorphic leaf-like structures in place of awns, and are controlled by two loci, *lel1* and *lel2*, with additive effects [15]. *lel1* has been linked to SNP markers 1_0876 and 2_0943 on 2H [3], about 6.1 cM distal from the molecular marker MWG733A [19]. *lel2*, also known as *lks5* on 4H [3], is required for expression of *leafy lemma* phenotypes [19]. The chromosome location of the short awn locus *lks6* remains to be determined [45], as inconsistent mapping data have linked it with markers from 1H, 5H, and 6H [3].

*Lsa1* (awnless on lateral spikelets), a unique locus specifically regulating awn development on lateral spikelets, has recently been identified in F_2_ of several barley crosses [44]. In these F_2_ populations, two-rowed plants with awned lateral spikelets and six-rowed plants with awnless lateral spikelets have been isolated. *Lsa1* is different from *vrs1/lr*, because recombination between *Lsa1* and *vrs1* on 2H has been observed [44]. Moreover, the dominant state of *Lsa1* represses awn development on the lateral spikelets, but it is the recessive state of *lr/vrs1* that inhibits lateral spikelet awn formation [29]. There are three dominant awnless genes, *B1*, *B2* and *Hd* (hooded) in wheat, among which *B1* acts as a dominant suppressor of the hooded phenotype [46]. Similarly, barley *Lsa1* also exhibits dominant epistatic effects on the hooded allele *Kap1.* The homolog of wheat *B1* has been identified as *HvFT3* in barley [47], which has a similar chromosome location as *Lsa1*, suggesting that the counterpart of wheat *B1* in barley could be a candidate for barley *Lsa1* [44].

*Lks1* (awnless) was once considered to be the same as *vrs1*, since they are closely linked on 2HL. This linkage has also complicated the fine mapping and gene cloning of *Lks1*. The awn inhibitor gene *B1*, which is dominant and inhibits awn development, has recently been cloned in wheat by several independent groups [48,49,50,51,52]. The wheat *B1* gene encodes a putative C2H2 zinc-finger protein with an EAR domain that is characteristic of transcriptional repressors. The cloning of wheat *B1* may facilitate cloning of barley *Lks1*, because the synteny of their locations on chromosomes between barley and wheat may provide useful hints for gene cloning.

The length of awns is controlled by the interactions of several awn genes, including *lr1* (for reduced length of lateral awns), *lks5* (or *lk5*, for short awn 5) and *lks2* (or *lk* and *lk4*, for short awns [53,54]. The genetic effect of *Lr1* is more prominent than that of *Lks5*, while *Lks2* and *Lks5* act additively to regulate awn length [44,53,54]. Thus, the interactions among these three awn genes play critical roles in the determination of awn length in barley.

More awnness loci have recently been identified as more DNA markers have become available. A quantitative trait locus (QTL), *qAL7.1*, located on 7HL, has been shown to regulate awn length [55]. A short awn length locus, *q**AL6.1*, has been mapped to 6HS with linkages to SSR marker HVM11 [56]. A new QTL, *qAL/SL/GD3.1*, for regulating awn length (AL), spike length (SL) and grain density (GD), has been localized on 3HL between SSR markers GBM1495 and HVM33 [57]. This QTL is not allelic to the other short awn locus *ari-a*, which is located on 3HS [3]. A QTL on 3H, *qAL3.1* for conferring short awn, dwarfing and short spike, has been mapped to a location of 3H very close to the dwarf gene *uzu1*, the DArT marker bPb-0079 and the SSR marker HVM44 [58]. Another awn length QTL, *qAL3.2*, has been identified on 3H near the centromere and between the SSR markers Bmac0067 and Bmag0006 [59].

By screening for suppressors (*suK*) of the *Kap1* (*Hooded*) mutation, five complementation groups, *suKB-4*, *suKC-33*, *suKD-25*, *suKE-74* and *suKF-76*, have been isolated and they all display a short-awn phenotype [60]. Among them, *suK B*, *C*, *E* and *F* loci map to a short interval of chromosome 7H, close to the location of another short-awn mutant *lks2*. Allelism tests suggest that these *suK* loci are different from *lks2*. The *suK D* locus maps to 5H between linkage subgroups 66 and 67 [60].

In addition to the *sbk1* locus on 2H, four *calcaroides* (*cal*) loci have been found to display similar subjacent hooded awn phenotypes. The mutant name *calcaroides* is derived from *calcar* in Latin, referring to the distinct “spur”- or “heel”-like structure that connects the mutant awn to the lemma tip (Figure 2O) [19,61]. *cal 23* is mapped to chromosome 7H between markers e3432-2 and e4040-3, while *cal d4* is mapped to 3H between e4146-4 and e3633-7. The dominant *cal C15* and the recessive *cal b19* are mapped to two tightly linked loci on chromosome 7H in a location about 4.2 cM distal to marker e4040-4 [19].

Besides the short awn loci *lks2* and *lks5*, there is a group of *ari* mutants, named after *breviaristatum* in Latin for short awns and a set of *brh* mutants (*brh1*–*brh18*), named after *brachytic* for dwarfism. The *brh* mutants have a short seedling leaf, reduced culm length and short awns. *brh16* is found to be allelic to *ari-o* on 7HL. Of the 140 *ari* mutant isolates tested for allelism, 17 loci each have multiple allelic mutants and two loci are represented each by a single allele [41]. *ari-a* and *ari-d* (*lks2*) have been placed on chromosomes (Figure 3). Some of the *ari* awn mutants are known to be involved in brassinosteroid (BR) biosynthesis or signaling, inhibiting the growth of the whole plant, including its awns. Besides displaying the short awn phenotype, the *brh* mutants are also shorter in height than the control [15]. *brh3*, encoding the barley brassinosteroid-6-oxidase, is involved in the BR biosynthesis pathway. *brh13*, encoding the barley C-23α-hydroxylase cytochrome, shows a BR deficient phenotype, including reduced culm length and short awns [15]. In order to differentiate the defective BR-signaling from the impaired BR-biosynthesis mutants, brassinolide, the most active BR, is applied exogenously to dark-grown seedling leaves that exhibit a characteristic rolling phenotype [62]. Using this leaf-unrolling bioassay, short awn mutants *ari.256*, *ert-ii79* and *uzu1.a* are found to have no response to brassinolide treatment, suggesting that they are defective in BR-signaling. Sequencing of the *HvBRI1* gene in these mutants reveals that mutations in *ari.256* and *ert-ii79* demolish the BR-binding site of the receptor. Other *ari* mutants, including *ari-u.245* and *ari-o.40*, are found to respond to brassinolide treatment, suggesting that they are impaired in the BR-biosynthesis pathway [62].

Most of these awnness loci were initially identified from studies using the classical Mendel genetics approach, which allows the placement of awn loci on linkage maps, but often leads to imprecise mapping locations on chromosomes and misidentification of the same loci as different genes conducted by different groups. Results obtained from these historic observations, though useful and interesting, have limited practical value in gene cloning. Today, the completion of the barley genome sequencing project has offered enormous information and advantages for molecular marker-assisted mapping of awn loci. Using the available barley genome platform, fine mapping of various awn loci identified in historic studies and elimination of redundant loci reported by different groups have become possible and convenient. 

Of the 12 awnness loci, both the awnless gene *Lks1* and the hooded awn gene *Kap1* or *sbk1* may become important targets for feed barley improvements. Because the awn is an undesirable trait in feed barley, it has been proposed that the awnless gene *Lks1* could be introduced to feed barley cultivars. However, this strategy is currently met with the challenge that the available *Lks1* mutant is associated with reduced grain yield. The *Kap1* gene appears to be a more suitable target gene for feed barley, because its mutant does not exhibit reductions in grain yield. In the USA, new barley cultivars containing *Lks1* or *Kap1* have been released for forage uses. If the reduction of grain yield cannot be resolved in awnless barley cultivars, one of the alternative approaches is to target short awn or hooded awn genes for improvement in forage breeding programs.

## 6. Interactions of Awnness Genes

Historic studies have provided several lines of evidence for the interactions between awnness genes in barley [32]. Awnness phenotypes are controlled by different awnness loci that interact with each other [63]. Analysis of the interactions among them can potentially reveal the mechanisms by which awn development is regulated. Awnness gene interactions have been systematically studied recently [44,63]. Analysis of the interaction between the hooded gene *Kap1* (*K*) and the short awn gene *lks2* has revealed that *lks2* is recessive and epistatic to the hooded gene *K/k* [32]. A new gene *Lsa1*, dominating the awlessness on lateral spikelets, was identified when studying the interactions of awnness loci [44]. Different from the above recessive epistasis of awned to hooded, multiple dominant epistasises of awnless to hooded and hooded to awed have been identified recently, and there are overlapping effects of the awnless genes on the central and lateral rows [44]. There are three dominant awnless genes (*B1*, *B2* and *Hd* (hooded)) in [46]. It is interesting to know that *B1* acts as a dominant suppressor of the hooded phenotype in wheat, the same as *Lsa1* to the hooded allele *Kap1* in barley; the *B1* homolog in barley *HvFT-3* has similar location to *Lsa1*, suggesting that *Lsa1* could possibly be *B1* [44].

## 7. Pleiotropism of Awn Genes

Genetic pleiotropism describes the effect of a single gene on multiple, unrelated traits. The mutation of a pleiotropic gene may thus lead to changes in several seemingly unrelated attributes of an organism. For mapping analysis, it is critical to distinguish the pleiotropic effect of one gene on multiple traits from a close linkage of two independent loci. For a long time, it has been believed that the awnless gene *Lks1* on chromosome 2H may be allelic to the row-type locus *vrs1*, and that the awnless phenotype and the six-row spike trait were just the result of pleiotropic effects of the same locus. This assumption was made on the basis of the lack of recombination between the two loci when a line with a dominant short awn gene *Lks1* and the two-rowed gene *Vrs1* is crossed with another line having long awns and six-rowed spikes (*lks1/lks1 vrs1/vrs1*) [43,64,65]. However, further genetic analysis has suggested that the two loci are independent but closely linked and lie near a paracentric chromosome inversion that may inhibit recombination between them [66].

The pleiotropic effects of short awn genes are especially common. A QTL, *qAL/SL/GD3.1*, has been shown to control all three traits of grain density, spike length and awn length [57]. The dwarf gene *uzu1* has pleiotropic effects on the length of leaves, culms, rachis internodes, awns, glumes and kernels [33]. A dwarfing gene closely linked to *uzu1* on chromosome 3H, is pleiotropic for short spikes, short awns and high grain density [58]. Allelism tests show that *lks2*, which reduces the awn length to about one half of the normal length (*Lks2*), is allelic to the unbranched style 4 (*ubs4*), which also displays sparse stigma hairs [12]. The short awn gene *lks5* on chromosome 4H has pleiotropic effects on the spikelets and produces extra florets [67].

Understanding the pleiotropism of awn genes will facilitate elucidation of how they function in awn development. Most awned varieties in barley and wheat have higher yields than awnless ones because of the grain-filling function of awns in photosynthesis [68,69]. Despite the overall higher potential of awned cultivars, there are awnless or short awned varieties that have good performance in yield in some ecological areas [13]. This unexpected high yield of half-awned or awnless varieties has been attributed to the high kernel number per spike, not to grain size [70]. Awnless or short awn traits normally decrease the size of grains but increase the number of spikelets/florets per spike [70]. However, the awned trait usually reduces the grain number and increases the grain size and harvestable yield [13,71]. The wheat awnless gene *B1* also decreases kernel weight, kernel length and test weight, but increases the number of spikelets per spike [49]. Overexpression of the barley *HvFT3* accelerates the initiation of spikelet primordia [48], and *HvFT3* is believed to act upstream of the row type genes *vrs1*, *vrs4* and *int-c* [47]. Therefore, awn genes not only regulate awn development, but also play important roles in other physiological and developmental processes such as grain filling and specification of spikelet initiation and row types.

## 8. Concluding Remarks

Awns are a bristle-like appendage that is formed on the tip of the lemma. They protrude upward from the canopy layer and are fully exposed to the sunlight. They are highly active in photosynthesis and contribute significantly to the filling of developing grains. Barley awns are highly diverse in morphology, ranging from long to short or awnless types, and from straight to hooded or crooked shapes. The morphological diversity, importance to grain yield and ease of study of barley awns have intrigued plant geneticists for several decades. A large set of genetic loci associated with the development of awns have been identified genetically and mapped to chromosomes. A few of them have recently been cloned and characterized. As compared with rice or maize, whose genome sequencing projects were completed much earlier than that of barley, there are research gaps that need to be addressed in barley. More awnness loci need to be subjected to fine mapping and then to gene cloning. Further characterization of the cloned or mapped awnness genes at the molecular biology, biochemistry, cell and developmental levels is a challenging task facing a whole generation of barley molecular biologists worldwide. On the other hand, barley has more plentiful awn mutants than other cereal crops and is an excellent model crop for awn studies. The availability of the Bowman backcross lines, which produce mutant phenotypes with the same genetic background, is one example of why more plant biologists should realign their research focus on barley.

The recent completion of the barley genome sequencing project, together with the application of novel sequencing and bioinformatic technologies, will greatly facilitate fine mapping and gene cloning of awn loci in barley. More and more awn genes will soon be cloned and characterized at the molecular biology and biochemistry levels. A better understanding of the molecular nature of awn genes would pave a solid foundation for further investigations on gene–gene interactions and molecular mechanisms of awn and grain development. These studies would eventually benefit barley breeders in developing new cultivars with improved yield and adaptability to various environmental conditions.

## Figures and Tables

**Figure 1 genes-12-01613-f001:**
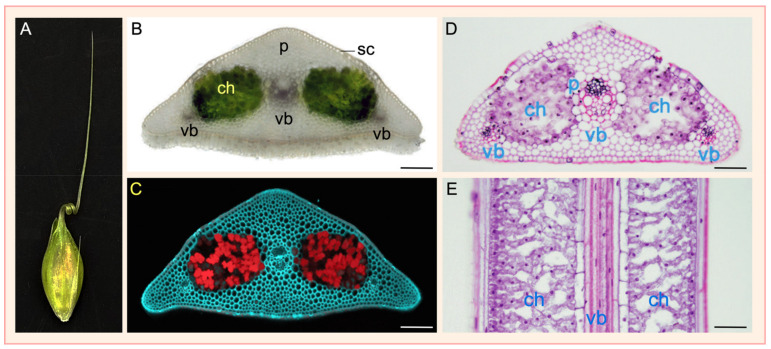
Anatomical structures of barley awns. (**A**), The awn is a characteristic floret organ and an extension of the lemma of spikelets in barley. Shown is a twisted awn of the barley *ari-a* mutant grown in the green house of the University of Idaho, photo taken by B.H. (**B**,**C**), A transverse section of a typical barley awn. The chlorenchyma cells (ch) are green under transmission light microscopy due to the presence of chlorophylls (**B**) but are red under a fluorescent microscope because of the autofluorescence of chlorophylls (**C**). (**D**,**E**), Transverse and longitudinal awn sections of barley cultivar Bowman under a light microscope after staining with Safranin O, a red dye for nuclei and cell walls. On a longitudinal awn section, the chlorenchyma cells are organized as two long strands that are placed among the three vascular bundles (vb). The chlorenchyma cells are active in photosynthesis and provide a major contribution of carbon sources for grain filling. p, parenchyma cells; sc, sclerenchyma cells. Bars, 100 μm. Images of (**B**,**C**) are taken from [11] and (**D**,**E**) from [12], with permissions from publishers.

**Figure 2 genes-12-01613-f002:**
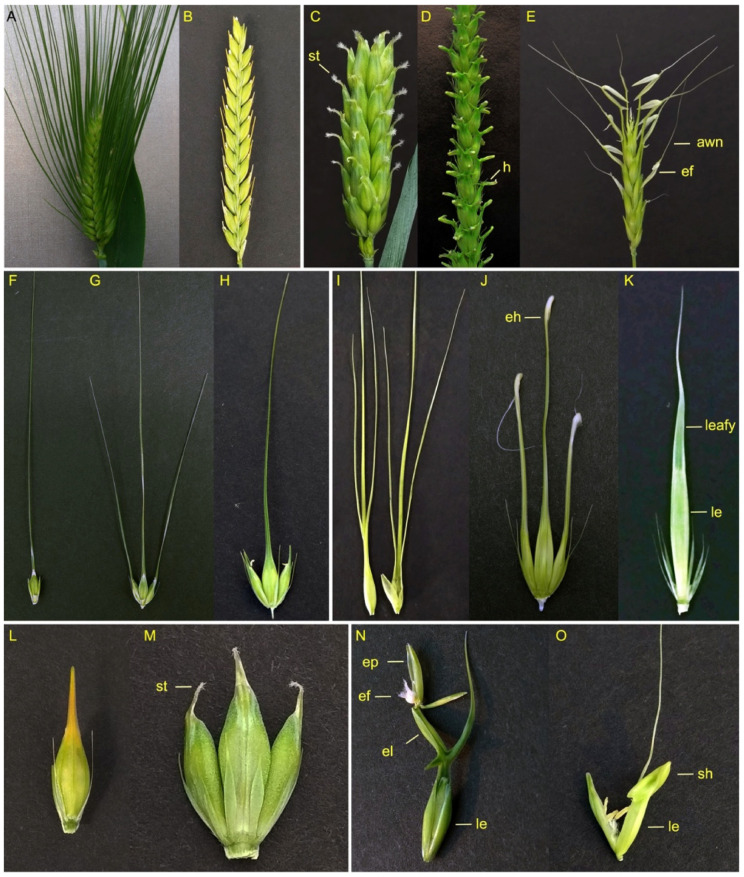
Morphological diversity of awns in barley cultivars and mutants. (**A**), 6-rowed spike with long awns. (**B**), awnless 2-rowed spike. (**C**), 6-rowed spike with stigma-like short, crooked awns. (**D**), Hooded spike. (**E**), Hooded spike with long awns. (**F**,**G**), 2-rowed and 6-rowed spikes with long awns. (**H**), 6-rowed spike having long awns on the central spikelet and awnless lateral spikelets. (**I**), Front- and side-views of a triple awn. (**J**), 6-rowed spikelets with elevated hoods. (**K**), spikelet with leafy awn. (**L**), awnless spikelet. (**M**), 6-rowed spikelets with stigma-like short, crooked awns. (**N**), hooded floret with an extra floret. (**O**), subjacent hooded floret. st, stigma-like short, crooked awn; h, hooded awn; ef, extra floret; eh, elevated hooded; leafy, leafy awn; le, lemma; ep, extra palea, el, extra lemma.

**Figure 3 genes-12-01613-f003:**
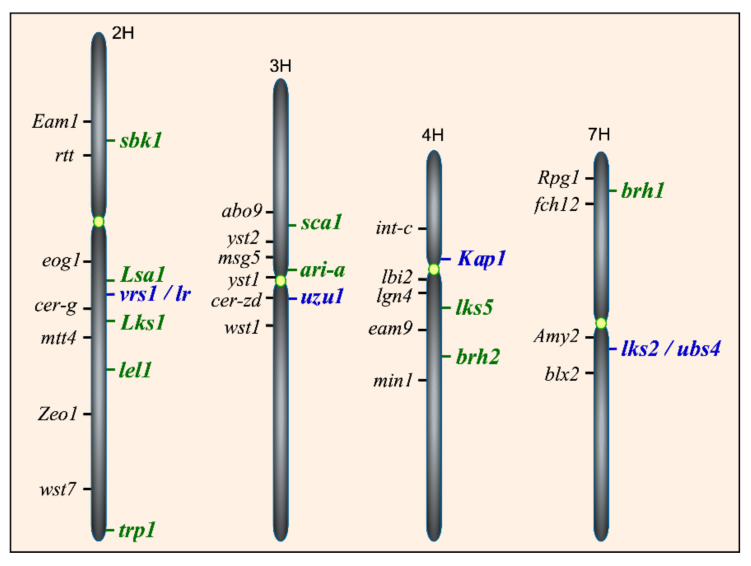
Cloned and mapped awn loci on barley chromosomes. Chromosomes 1H, 5H, and 6H do not contain known loci for awns and are not shown. Illustrated on the right sides of chromosomes are four cloned genes (shown in blue, *vrs1/lr*, *uzu1*, Kap1 and *lks2/ubs4*) that regulate awn development and 10 mapped awnness loci (shown in green) that are described in this review. Morphological markers near the awnness loci are indicated in black and listed on the left side of chromosomes.

**Table 1 genes-12-01613-t001:** Cloned awnness genes in barley.

**Gene Names**	*Kap1* (*hooded lemma 1*)	*vrs1* (*six-rowed spike 1*)	*lks2* (*short awn 2*)
**Other Names**	*K*, *HvKnox3*	*lr*, *HvHox1*	*lk2*, *lk4*, *ari-d, ubs4*
**Awn Phenotype**	Hooded lemma	Reduced lateral spikelet appendage (*lr*)	Short awn
**Chromosome**	4HS	2HL	7HL
**Cloning Method**	Homologous cloning	Positional cloning	Positional cloning
**Mutation Types**	Insertion in intron 4	Missense, nonsense, and splicing site changes	Missense mutation
**Protein Length ^a^**	364 aa	222 aa	344 aa
**Protein Type**	KNOX family transcription factor	HD-ZIP family transcription factor	SRS family transcription factor
**Orthologs**	Rice *OSH15* and maize *KNOTTED-1*	OsHox12 and OsHox14 in rice	Os06g0712600 in rice
**References**	[22,24]	[25,26]	[12,27]

^a^ aa, amino acid residues.

**Table 2 genes-12-01613-t002:** Mapped genetic loci for awnness in barley.

Locus	Other Names	Awn Phenotypes	Chr ^a^	Ref. ^b^
*sbk1*	*sk, cal-a*	Subjacent hood	2HS	[19,37]
*lel1*	*lel*	Leafy lemma	2HL	[3,38]
*trp1*	*tr*	Triple awned lemma	2HL	[24,39]
*sca1*	*sca*	Short, crooked awn	3HS	[40]
*lks5*	*lk5, ari-c, lel2*	Short awn	4HS	[27,39]
*ari-a*	*lk7*	Short awn	3HS	[41,42]
*lks6*	*Lks.q*	Short awn	ND ^c^	[3]
*Lks1*	*Lk*	Awnless	2HL	[43]
*Lsa1*	*Lsa*	Awnless on lateral rows	2H	[44]

^a^ Location of the short (S) and long (L) arms of chromosomes. ^b^ Reference (ref.) sources. ^c^ ND: chromosome location not determined.

## Data Availability

Not applicable.

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
