# Peer review of "Genetic Loci Underlying Awn Morphology in Barley"

_genes, 2021, doi:10.3390/genes12101613_

Round 1

Reviewer 1 Report

The manuscript entitled “Genetic loci underlying awn morphology in barley” is reviewed for the genomic loci of a specific morphological character “awn”. I appreciate author(s) to review for the genetic loci of a specific characters.

However, I don't find the importance of this review. I don't understand the reason why the author(s) choose to review this topic.What was the specific objective(s) to review?

The identification of genetic loci in last two-decade (if we consider from this review, such as Table 2), only few (5-6) articles were focused on “awn” genomic loci. It is not sufficient to choose a review article.

Figure 3: Now a days, the reliability of the identified loci for any character is a major question? Why do not authors discuss about the reliability of the identified loci? Are there any specific loci that has been repeatedly detected by several researcher?

Is there any linkage between cloned (vrs1/lr) and nearest mapped (Lsa1 or Lks1) awn loci in Chromosome 2H?

It has been summarized that there are 12 loci has been identified for awn in barley, but which loci can be considered as the major loci. What can be impact of the identified loci for the further improvement awn characters in barley? Author(s) should discuss and give us a clear idea of the past investigations and future.

This review article does not consist any critical analysis of the previous data, it has only the compilation of the previous data.

Authors should not forget the important context of any typical review article such as (i) advancement, (ii) research gap that may lead to high impact, (iii) mystery to be solved etc.

[Page 8, line 223-233] I think, you can use Figure 3 to visualize QTLs.

I think authors should reconsider and re-write the manuscript.

There are some minor points are also need to consider such as

[Page 2, line 66-70] Check the levels (A, B, C, etc.) of Figure 1.

There are many citations that are very old. I think authors should skip the least important old citations.

Reviewer 2 Report

The manuscript is well written.

Lines 169-170: The authors say" Awnness genes with functions other than transcription factors will be uncovered, which will p..." . How can the authors be so sure about the nature of awnness genes, they should change "will" to 'may be'

Author Response

Comments and suggestions of Reviewer 2

The manuscript is well written.

Lines 169-170: The authors say" Awnness genes with functions other than transcription factors will be uncovered, which will p..." . How can the authors be so sure about the nature of awnness genes, they should change "will" to 'may be'

Re: We appreciate the comments by this Reviewer and have accordingly changed “will” into “may be” in line 173.

Round 2

Reviewer 1 Report

I am sorry to say that, I do not find any improvement in this manuscript.
On the other hand, responses by the author(s) for the questions arose are not satisfactory in all every points such as the specific objective(s) to review,  impact of the identified genomic loci for the further crop improvement, research gap that need to be concerned for the precious information etc.?

I have mentioned previously that authors fails to critically analyze the previous information.
